# Preventing Sexual Violence and Strengthening Post-Victimization Support Among Adolescents and Young People in Kenya: An INSPIRE-Aligned Analysis of the 2019 Violence Against Children Survey (VACS)

**DOI:** 10.3390/ijerph22060863

**Published:** 2025-05-30

**Authors:** Denis Okova, Akim Tafadzwa Lukwa, Olufunke A. Alaba

**Affiliations:** 1Health Economics Unit, School of Public Health, Faculty of Health Sciences, University of Cape Town, Cape Town 7700, South Africa or denisokova@gmail.com (D.O.); or tafadzwalukwa@gmail.com (A.T.L.); 2Division of Family Medicine, Family, Community and Emergency Care (FaCE), Faculty of Health Sciences, University of Cape Town, Cape Town 7700, South Africa; 3Partnership for Economic Policy, Nairobi 00100, Kenya

**Keywords:** sexual violence, help-seeking, disclosure, adolescent girls and young women (AGYW), adolescent boys and young men (ABYM)

## Abstract

Background: Sexual violence against adolescents and young people (AYP) remains a public health concern. This study explores patterns of sexual violence and help-seeking behaviour as well as their associated risk/protective factors with guidance of a technical package (INSPIRE) designed to reduce sexual violence in low-resource settings. Methods: The 2019 Violence Against Children Survey (VACS) dataset comprises 788 males and 1344 females. After describing the prevalence and patterns of sexual violence and help-seeking behaviour (informal disclosure, knowledge of where to seek formal help, seeking formal help, and receipt of formal help) among 13- to 24-year-old AYP, logistic regression models were then fitted to predict past-year sexual violence and informal disclosure among adolescent girls and young women (AGYW). Results: More young women than young men informally disclosed sexual violence experience (46% versus 23%). Gender inequitable attitudes [AOR 3.07 (1.10–8.56); *p* = 0.03], experiencing emotional violence at home [AOR 2.11 (1.17–3.81); *p* = 0.01] and cyberbullying [AOR 5.90 (2.83–12.29); *p* = 0.00] were identified as risk factors for sexual violence among AGYW. Life skills training [AOR 0.22 (0.07–0.73); *p* = 0.01] and positive parental monitoring [AOR 0.31 (0.10–0.99); *p* = 0.05] were found to be protective against sexual violence among AGYW. Positive parental monitoring [AOR 3.85 (1.56–9.46); *p* = 0.00] was associated with an increased likelihood of informal disclosure among AGYW. Conclusions: As Kenya intensifies efforts towards sexual violence prevention, this study underscores the need to develop and strengthen policies and programs on life skills training, cultural norms, and positive parenting, as well as improve awareness and access to post-violence response and support services.

## 1. Introduction

Violence (physical, emotional or sexual violence) against young people is a global social, public health and human right concern and a development priority [1,2]. Global estimates show about 50% (1 billion) of children in the world experience some form of violence each year [3,4,5]. However, it has been argued that the reported violence statistics are largely under reported and under-acknowledged in developing countries (hide in plain sight) [6]. This has been attributed to the normalization of violence [7], cultural norms and societal expectations discouraging open discussion of violence [8], limited research and awareness [9], and underinvestment in social services and preventive programs [4] among other factors. A six-country study (Cambodia, Haiti, Kenya, Malawi, Nigeria and Tanzania) on childhood violence reported lifetime prevalence rates for sexual violence ranging between 6 and 36% and between 50 and 84% for physical violence among children aged 13 to 17 years. For Kenya, the lifetime prevalence rates for sexual violence were 13% among males and 24% among females [10]. As of 2019, it was estimated that 5 in every 10 children experienced or witnessed physical, sexual, or emotional violence, while 1 in every 4 children experienced sexual violence [11]. Across select countries in sub-Saharan Africa, the prevalence rates of forced sex/rape among girls aged 15 to 19 years in Zimbabwe, Malawi, Rwanda, Tanzania, Zambia, and Kenya were 41%, 38%, 29%, 27%, 26%, and 24%, respectively [11].

Sexual violence among adolescents and young people (AYP) has multiple risk factors. These risk factors include but are not limited to prior victimization of a child or their family member, concurrent forms of abuse in the child’s environment, parental problems, parenting problems, non-nuclear family structure, child problems like mental health issues, and being female [12,13]. Violence, including sexual violence, has long-lasting and costly social and emotional effects [4]. For instance, there exists a strong association between sexual violence and sexually transmitted infections and HIV [14,15,16,17]. Additionally, individuals exposed to violence are also more likely to engage in risky behaviours such as having multiple sexual partners, early sexual debut, erratic condom use, and drug or alcohol abuse [16,18,19]. Sexual violence is also associated with profound psychological effects, including increased risk of depression, anxiety, and post-traumatic stress disorder (PTSD) [20,21].

Without doubt, sexual violence among AYP is a concern for countries with HIV epidemics like Kenya. Kenya for instance has one of the largest HIV epidemics globally, with 1.4 million people living with HIV (PLHIV) [22], and with over 60% of the total adult new infections occurring among AYP aged 15–29 years, as per the 2020/21–2024/25 Kenya AIDS Strategic Framework II [23]. For this reason, such countries have adopted violence prevention strategies as one of the means to fight the HIV epidemic. In 2015, the Kenyan government in conjunction with the U.S. President’s Emergency Plan for AIDS Relief (PEPFAR) launched the DREAMS (Determined, Resilient, Empowered, AIDS-free, Mentored, and Safe) program targeting adolescent girls and young women (AGYW) living in high-HIV prevalence areas. The initiative aimed at lowering the risk of sexual violence and HIV through education subsidies, HIV and violence prevention education, and financial literacy education [24,25].

As a result of limited nationally representative data on violence to inform policy and programming, the Centers for Disease Control (CDC) and other development partners started supporting global south countries in 2007 to roll out the Violence Against Children Surveys (VACS) to fill this evidence gap. VACS have been implemented in 22 African countries, including Kenya. Kenya has conducted two Violence Against Children Surveys (VACS), the first-ever in 2010 and the second one in 2019. These two datasets are the only nationally and sub-nationally representative statistics available on violence [26].

To further consolidate efforts towards moving closer to achieving Sustainable Development Goal (SDG) 16.2 (To end abuse, exploitation, trafficking and all forms of violence against and torture of children), the World Health Organization in collaboration with development partners launched the INSPIRE technical package in 2016. INSPIRE comprises seven evidence-based strategies for preventing (sexual) violence among children in low-resource settings. They are **I**mplementation and enforcement of laws, **N**orms and values, **S**afe Environments, **P**arent and caregiver support, **I**ncome and economic strengthening, **R**esponse and support services, and **E**ducation and life skills [27]. Simultaneous and continuous targeting of these intervention points is predicted to reduce the incidence of violence against children and young people. Since the release of this technical package in 2016, at least 67 countries, including Kenya, have demonstrated some level of engagement with the framework and adoption of the proposed strategies [1]. Further details on each of the INSPIRE strategies and how Kenya is implementing the framework are provided in Appendix A.

Kenya is making progress in reducing sexual violence rates and improving response and support services for victims. However, as of 2019, the prevalence rate of childhood sexual violence was 16% for females and 6% for males. Three out of five women who experienced sexual violence reported multiple incidents before the age of 18, while 90% of females aged 18 to 24 years who experienced sexual violence did not seek help [28]. With respect to disclosure, as per the 2019 VACS, less than 50% of AGYW and less than a quarter of adolescent boys and young men (ABYM) disclosed experiences of sexual violence [28]. In Kenya, survivors of sexual violence may pursue justice through the formal criminal system (focused on prosecution) or traditional mechanisms led by community elders, which emphasize reconciliation and may include compensation to the survivor’s family. These parallel systems can conflict, and many cases go unreported due to a preference for traditional approaches perceived to offer more tangible benefits than the criminal justice system [29]. Additionally, Kenya’s social and cultural norms of victim-shaming further silence them [30]. While the provision of comprehensive post-rape services is improving, there is still much to be desired in rural areas where facilities are ill equipped and service providers undertrained [31].

As the Kenyan government scales up the uptake of INSPIRE strategies for (sexual) violence prevention (Appendix A), there is need to test these interventions feasibly and affordably using observational data like VACS to investigate associations between INSPIRE-aligned risk or protective factors and outcomes like sexual violence experience and disclosure. Such an analysis could guide the country’s future sexual violence policy and/or programme selection. A similar INSPIRE-aligned analysis has been conducted in South Africa by Cluver et al. (2020) [32] to understand violence prevention accelerators for children and adolescents using two pooled cohorts. Cluver et al. found that a combination of interventions termed “violence prevention accelerators”, including caregiver monitoring, food security, school access, and parenting support, significantly reduced sexual violence victimization among adolescents in South Africa. Their path analysis also revealed that the cumulative effect of these interventions was greater than any single measure alone [32].

Therefore, this study examined (1) the patterns of lifetime sexual violence experiences and help-seeking behaviour (informal disclosure, knowledge of where to seek formal help, seeking formal help, and receipt of formal help) among AGYW and ABYM, (2) the associations between INSPIRE-aligned risk/protective factors and past-year sexual violence among AGYW, and (3) the associations between INSPIRE aligned risk/protective factors and informal disclosure of lifetime sexual violence experiences among AGYW. In this study, informal disclosure refers to the act of sharing one’s experience of sexual violence with non-institutional sources such as friends, family members, or other trusted individuals, rather than with formal authorities like healthcare providers, police, or social workers.

Gender is a critical determinant of both exposure to sexual violence and patterns of help-seeking [33]. Evidence consistently shows that AGYW experience higher rates of sexual violence [10], while ABYM often face stigma and social silence when victimized [34,35]. These gendered dynamics shape not only who is at risk, but also who seeks help and how that help is received. This study positions gender as a central analytical lens to better understand how social norms, risk factors, and response mechanisms operate differently across groups.

## 2. Materials and Methods

### 2.1. Data

This study used the 2019 Kenya Violence Against Children Survey (VACS) data. VACS are nationally representative, cross-sectional household surveys targeting non-institutionalized females and males aged 13 to 24 years, focusing on physical, emotional, and sexual violence experienced in the past year and lifetime, as well as risk and protective factors, health outcomes, and service access. Details on these variables are provided in Appendix B. The study had 788 males and 1344 females. This survey used a three-stage cluster sampling design with random selection of enumeration areas as the first stage, households as the second stage, and an eligible participant from the selected household as the third stage. The sampling was conducted without replacement. Importantly, separate Enumeration Areas (EAs) were randomly selected for males and females, with more EAs allocated to females (155 vs. 109) to ensure sufficient representation. Data collection was conducted through interviewer-administered face-to-face surveys using trained interviewers, with gender-matching of interviewers and respondents to ensure comfort and confidentiality [28].

### 2.2. Ethical Considerations

To ensure the protection of the welfare and rights of human research participants, the Kenya 2019 VACS survey protocol was reviewed and approved by the Kenyatta National Hospital/University of Nairobi ethics review committee and the Centre for Disease Prevention and Control (CDC), the University of California San Francisco (UCSF), and the Population Council institutional review boards. This secondary analysis used publicly available Violence Against Children (VACS) datasets. Even so, ethics approval was obtained from the Human Research Ethics Committee (HREC) at the University of Cape Town (HREC REF: 282/2022).

### 2.3. Measures

#### 2.3.1. Outcome Variables

The outcome variables for this study were sexual violence experiences and post-violence help-seeking behaviours. Sexual violence experience was defined as self-reported occurrences of unwanted sexual touching, attempted forced sex, physically forced sex, or pressured sex, either in the past twelve months or over a lifetime. Exposure to any of these forms was recorded as a sexual violence experience. Post-violence help-seeking behaviours were categorized into four domains: informal disclosure, knowledge of where to seek formal help, seeking formal help, and receipt of formal help. For informal disclosure, respondents who had experienced unwanted sexual experiences were asked if they had told anyone (parent, friend, sibling, any other relative, a neighbour) about these incidents. Knowledge of formal help sources was assessed by asking respondents if they knew of a hospital/clinic, police station, helpline, social welfare office, or legal office where they could seek help after experiencing sexual violence. Seeking formal help was measured by asking if respondents had sought help from any of these formal sources. Seeking formal help would have been a proxy for formal disclosure, but this was not pursued due to sample size limitations. Finally, respondents who sought help were asked if they received assistance from these formal sources, thus assessing receipt of formal help. All these outcome variables were binary. Further detail on how these variables were operationalized is shown in Appendix B.

#### 2.3.2. INSPIRE-Aligned Predictor Variables

The 2019 VACS was conducted after the INSPIRE framework had been published. This framework informed the choice of predictor variables. We picked these variables focusing on three arms of the INSPIRE framework: **N**orms and Values, **P**arent and Caregiver support, and **E**ducation and life skills. Guided by a review of the literature [13,32,36,37,38,39], these predictor variables were positive parental monitoring, life skills training, cyberbullying, experience of physical violence at home, experience of emotional violence at home, and gender norms (endorsement of gender inequitable attitudes and acceptability of wife-beating/Intimate Partner Violence). These variables were picked for their availability in the dataset. Moreover, for these three arms and their corresponding variables, it was possible to reasonably attenuate concerns about temporal precedence bias, as they captured exposures likely to have occurred prior to the outcomes of interest. Only variables that were clearly measured as occurring prior to the outcome timeframe, such as childhood experiences, household environment, and caregiver behaviours, were included. For instance, when modelling past-year sexual violence or lifetime disclosure, predictors reflecting more distal or stable characteristics were prioritized. Variables with unclear or overlapping temporal reference points relative to the outcome (e.g., concurrent behaviours or retrospective perceptions) were excluded to minimize the risk of reverse causality and misclassification. All these predictor variables were binary. Further detail on how these variables were operationalized is shown in Appendix B.

#### 2.3.3. Covariates

Covariates were selected based on prior evidence of their association with experiences of sexual violence and help-seeking behaviours among adolescents [13,32,37]. The control variables considered in this analysis were age, marital or relationship status, orphanhood, education status, HIV testing, and household poverty. Age was a continuous variable, while the rest were binary variables. As the models were stratified by sex, a separate sex variable was not included in the adjusted models.

### 2.4. Data Analysis

All analyses for this study were conducted using STATA version 17 (Stata Corp. Inc., College Station, TX, USA). The analyses were weighted to account for the complex design of the VACS. We conducted gender-disaggregated analyses instead of controlling for gender as a covariate to reflect theoretical and empirical distinctions in how adolescent girls and boys experience and respond to sexual violence. Gender is a structural determinant that shapes risk exposure, disclosure norms, and access to services differently across groups [33,40]. Stratification allowed for identification of gender-specific associations with INSPIRE-aligned factors, which may be masked in pooled models. We first performed descriptive analyses to compute the prevalence and pattern of sexual violence (both lifetime and past-year events) and lifetime help-seeking behaviour for both sexes. Next, multivariate logistic regression models were fitted to predict past-year sexual violence and lifetime informal disclosure among females. As noted earlier, logistic models were not fitted for formal disclosure due to sample size limitations. From these models, adjusted odds ratios (aORs) and 95% Confidence Intervals (CIs) were reported. These models were not fitted for males due to sample size limitations.

## 3. Results

### 3.1. Descriptive Statistics

This study had 1344 female and 788 male participants (Table 1). A total of 20.9% of AGYW had ever been married or in a relationship, while 54.6% of them had at least secondary education. Among the same demographic, the prevalence of household poverty was 43.1%, while orphanhood by either a mother or a father or both was prevalent at 23.4%, and 20.8% of females reported having received life skills training in the past year. In terms of potential past-year sexual violence predictors, gender norms had the highest frequencies, with 61.8% of young women endorsing at least one gender inequitable attitude and 49.8% justifying wife beating (Intimate Partner Violence). With respect to violence at homes, 36.1% and 16.8% of AGYW reported experiencing physical violence and emotional violence at home, respectively. On sexual violence forms, lifetime violence forms had higher frequencies than past-year forms. But in both cases, attempted forced sex was the most prevalent form: 4% (past year) and 15.3% (lifetime). Among ABYM, 68.7% and 47.8% of them endorsed gender inequitable attitudes and wife beating, respectively (Table 1).

### 3.2. Pathway from Sexual Violence Experience to Help-Seeking

Table 2 as well as Figure 1, showing the pathway from lifetime sexual violence experience to help-seeking between males and females, reveal significant differences. A significantly higher proportion of females (25.2%; 95% CI: 22.4–28.3) than males (11.4%; 95% CI: 9.1–14.1) reported experiencing lifetime sexual violence (*p* = 0.00). Informal disclosure was also more common among females (45.1%; 95% CI: 37.1–53.4) compared to males (22.7%; 95% CI: 12.0–38.9), with the difference being statistically significant (*p* = 0.00). However, there were no statistically significant gender differences in knowledge of where to seek formal help (females: 33.7%, males: 33.1%; *p* = 0.85), seeking formal help (females: 11.3%, males: 6.8%; *p* = 0.25), or receiving formal help (females: 10.0%, males: 6.0%; *p* = 0.36). Notably, a significantly greater proportion of males did not disclose their experience to anyone (77.3% vs. 54.9%), highlighting more substantial barriers to informal disclosure among men (Figure 1).

### 3.3. Patterns of Past-Year Sexual Violence Forms Stratified by Predictor Variables

Table 3, which reports the patterns of sexual violence forms (past year) among AGYW by exposure variables, indicates several key findings. Those who endorsed gender inequitable attitudes reported high prevalence across all sexual violence forms: 78.4% (95% CI: 64.0–88.1) for unwanted touching, 61.8% (95% CI: 39.7–80.0) for attempted forced sex, 83.6% (95% CI: 51.4–96.1) for physically forced sex, and 82% (95% CI: 46.5–96.0) for pressured sex. Similarly, justification of wife beating was associated with significant prevalence rates: 48.8% (95% CI: 31.2–66.7) for unwanted touching, 70.9% (95% CI: 51.2–85.0) for attempted forced sex, 33.8% (95% CI: 9.1–72.2) for physically forced sex, and 24.8% (95% CI: 5.5–65.1) for pressured sex.

Among females who reported experiencing physical violence at home, the prevalence rates were 50.7% (95% CI: 35.0–66.3) for unwanted touching, 40.2% (95% CI: 22.1–61.6) for attempted forced sex, 73.8% (95% CI: 35.5–93.5) for physically forced sex, and 78.5% (95% CI: 39.6–95.3) for pressured sex. High prevalence across all sexual violence forms was also observed among females who had experienced emotional violence at home: 36.6% (95% CI: 23.5–52.0) for unwanted touching, 53% (95% CI: 32.9–72.1) for attempted forced sex, 21.4% (95% CI: 5.4–56.2) for physically forced sex, and 21.8% (95% CI: 5.1–58.9) for pressured sex.

Those who experienced cyberbullying had significantly high prevalence rates for all forms of sexual violence: 62.1% (95% CI: 44.6–76.9) for unwanted touching, 81.8% (95% CI: 65.0–91.6) for attempted forced sex, 85.3% (95% CI: 52.4–96.8) for physically forced sex, and 83.1% (95% CI: 49.6–96.1) for pressured sex. In contrast, females with positive parental monitoring reported lower prevalence rates: 12.1% (95% CI: 5.8–23.4) for unwanted touching, 14.3% (95% CI: 3.9–40.5) for attempted forced sex, and 7.3% (95% CI: 1.4–31.2) for physically forced sex, with no data reported for pressured sex. Lastly, life skills training appeared to have a protective effect, with prevalence rates of 19.7% (95% CI: 11.1–32.7) for unwanted touching, 5.6% (95% CI: 2.0–14.5) for attempted forced sex, 9.1% (95% CI: 1.5–40.2) for physically forced sex, and 8.2% (95% CI: 1.0–45.2) for pressured sex. These results highlight the significant impact of various predictor variables on the prevalence of different forms of sexual violence among adolescent girls and young women.

### 3.4. Associations Between Past-Year Sexual Violence Experience and INSPIRE-Aligned Predictors Among AGYW

Based on the logistic model predicting past year sexual violence experience among adolescent girls and young women (Table 4), several significant associations were identified. For every unit increase in age, the likelihood of females experiencing physically forced sex and pressured sex increased by 1.33 times (95% CI: 1.16–1.52; *p* = 0.00) and 2.17 times (95% CI: 1.80–2.62; *p* = 0.00), respectively. Girls who had ever been in relationships or married were 0.11 times (95% CI: 0.02–0.69; *p* = 0.02), 0.13 times (95% CI: 0.03–0.46; *p* = 0.00), and 0.31 times (95% CI: 0.14–0.67; *p* = 0.00) less likely to be victims of unwanted touching, pressured sex, or any form of sexual violence, respectively. Females who endorsed inequitable gender attitudes were 3.07 times (95% CI: 1.10–8.56; *p* = 0.03) more likely to experience unwanted sexual touching. Participants who received positive parental monitoring were 0.31 times (95% CI: 0.10–0.99; *p* = 0.05) and 0.20 times (95% CI: 0.06–0.70; *p* = 0.01) less likely to be subjected to unwanted touching and physically forced sex, respectively.

Females who experienced emotional violence at home were 2.11 times (95% CI: 1.17–3.81; *p* = 0.01) more likely to experience any form of sexual violence than those who did not. Those who had experienced cyberbullying were significantly more likely to be victims of various forms of sexual violence: 3.31 times (95% CI: 1.24–8.84; *p* = 0.02) for unwanted touching, 8.43 times (95% CI: 2.51–28.30; *p* = 0.00) for attempted forced sex, 5.84 times (95% CI: 1.05–32.49; *p* = 0.04) for physically forced sex, and 5.90 times (95% CI: 2.83–12.29; *p* = 0.00) for any form of sexual violence. Girls who had received life skills training on anger management and avoidance of physical fights and cyberbullying were 0.22 times (95% CI: 0.07–0.73; *p* = 0.01) and 0.49 times (95% CI: 0.26–0.92; *p* = 0.03) less likely to experience attempted forced sex and any form of sexual violence compared to those who had not received such training. These findings underscore the significant impact of various exposure variables on the likelihood of experiencing different forms of sexual violence among adolescent girls and young women.

### 3.5. Associations Between Informal Disclosure and INSPIRE-Aligned Predictors Among AGYW

Based on the logistic model predicting informal disclosure for lifetime sexual violence experiences among adolescent girls and young women (Table 5), several significant associations were identified. Adolescent girls and young women who received positive parental monitoring were almost four times more likely (aOR = 3.85, 95% CI: 1.56–9.46; *p* = 0.00) to informally disclose experiences of lifetime sexual violence than those who did not. Additionally, girls who experienced cyberbullying were more than twice as likely (aOR = 2.13, 95% CI: 1.31–3.45; *p* = 0.00) to informally disclose these experiences compared to those who had not been cyberbullied. This highlights the significant role that both positive parental monitoring and the experience of cyberbullying play in influencing the likelihood of informal disclosure among adolescent girls and young women who have experienced lifetime sexual violence. Other predictors such as endorsement of gender inequitable attitudes, justification of wife beating, and experiences of physical or emotional violence at home were not significantly associated with informal disclosure in this model.

## 4. Discussion

This study is a nationally representative cross-sectional study of Kenyan adolescents and youth aged 13–24 years, which mapped the patterns of sexual violence experiences and help-seeking behaviour, as well as how various INSPIRE-aligned risk/protective factors predict sexual violence and disclosure. The study found that the lifetime prevalence of sexual violence was 25.2% for AGYW and 11.4% for ABYM. This was consistent with other studies performed in low- and middle-income countries (LMICs), which showed that women were more likely than men to experience sexual violence [10,41]. It should also be noted that the prevalence rates of VACS 2019 in Kenya are lower than those found in VACS 2010 (36.2% in AGYW and 19.7% in ABYM) [36]. This decline in prevalence may be attributed to successful programmatic and policy interventions implemented between 2010 and 2019. These include criminalization of violence among young people, setting the minimum age of marriage at 18 years, and setting up child protection in select police stations across the country [28,31,36].

Endorsement of gender inequitable attitudes was significantly associated with unwanted sexual contact among AGYW. These findings support research by Jewkes et al. (2011) [42] in South Africa that unfair (male-favouring) views of gender relations enable sexual violence. This is explained by the social entrenchment in society of adversarial sexual beliefs in favour of men and hegemonic masculinity, both of which promote males’ sexual entitlement [43,44,45,46]. This is a reality check for governments given the global commitment by countries to make deliberate efforts towards achieving gender equality (SDG-5).

Another risk factor was experience of emotional violence at home. AGYW who experienced this were twice as likely to experience any form of sexual violence. This is consistent with other research on the role of domestic instability in enabling sexual violence [47,48,49]. Emotional violence, as one indicator of home instability, can lead adolescents and young people to leave their homes, seek tranquillity in places with poor supervision, and predispose to sexual violence [50].

Cyberbullying was also a significant risk factor for sexual violence. Young women who had been victims of cyberbullying were about six times more likely to be victims of any form of sexual violence. This is consistent with research on the role of cyberbullying as a mediator in the sexual violence cascade, where sexually abused youth are more likely to experience subsequent psychological victimization [51,52]. The cyberspace offers a perfect platform for psychological victimization. Alternatively, cyberbullying in schools can cause victims to engage in unhealthy coping mechanisms such missing school and/or engaging in drug or alcohol abuse, which predispose them sexual violence [53]. Moreover, considering that cyberbullying often includes sexually explicit harassment such as the non-consensual sharing of intimate images or coercive sexual messaging, these experiences may themselves constitute sexual violence [54,55,56]. This overlap suggests that cyberbullying can function both as a precursor to and a component of sexual violence, contributing to a continuum of abuse that warrants comprehensive prevention and intervention strategies [57]. Furthermore, the link between cyberbullying and sexual violence may reflect broader patterns of poly-victimization, where individuals experience multiple, co-occurring forms of abuse due to shared risk factors such as social marginalization, gender norms, and power imbalances [58,59]. Future research should explore whether victimization in digital spaces increases susceptibility to offline abuse, and whether perpetrators operate across domains [60]. Gender-sensitive analyses are especially needed to assess whether online harassment reinforces existing vulnerabilities among AGYW and ABYM. To advance this work, we propose future studies examine (1) cross-context perpetration patterns, (2) cumulative risk across digital and physical environments, and (3) the role of social support in mitigating harm while ensuring framing does not pathologize survivors.

Life skills training and positive parental monitoring were found to be protective against sexual violence in this study. AGYW who had received life skills training (anger management, violence and bullying avoidance) were less likely to experience any form of sexual violence. This was consistent with the findings of a randomized controlled trial of a life skills training program for males and females, *SteppingStones,* implemented in South Africa. This program, implemented among 15- to 26-year-old males and females to promote equal relationships between partners, communication, interpersonal skills, and assertiveness, was found to reduce instances of rape and intimate partner violence [61,62]. During the program’s implementation in Durban, South Africa, there was a reduction in the prevalence of sexual violence among females, a decrease in controlling behaviours by men within relationships, and improved gender attitude scores among both males and females [61]. Multiple studies in Kenya have also highlighted the protective nature of life skills training [63,64,65,66]. Two school-based life-skills training programs are present in Kenya: IMPower for AGYW and Moment of Truth for ABYM. These 6-week empowerment based behavioural interventions have been shown to help AGYW recognize and resist sexual violence, while also promoting positive, non-violent masculinities among ABYM [67]. Age-specific life skills training promotes communication, conflict management, and problem-solving skills, as well promotes healthy peer relationships. While this can be offered in schools as part of the curriculum, it can also be provided in informal settings within communities.

AGYW who received positive parental monitoring were less likely to be victims of unwanted touching and physically forced sex. This is line with research findings on the evaluation of positive parenting programmes such as the *Families Parents Matter Program* and *Parenting for Lifelong Health (PLH),* which showed that positive parenting may be protective against violence [37]. A recent INSPIRE-aligned analysis in South Africa to identify violence prevention accelerators among children and adolescents in South Africa also found that positive parenting and parental monitoring and supervision were associated with lower odds of sexual abuse [32]. Positive parenting directly protects children and young people from sexual violence through supervision, monitoring, involvement, and communication. Positive parenting, on the other hand, indirectly protects children by fostering their competence, well-being, and emotional self-efficacy, reducing their chances of becoming victims of sexual violence and increasing their ability to respond appropriately to abuse. A parenting program dubbed “Malezi Bora na Maisha Mazuri” (Good Parenting for a Good Life) has been implemented in Kenya targeting street connected mothers and has been found to reduce the use of corporal punishment among these parents, reduce parental stress, and increase positive parenting practices [68]. Scaling up such programs may be beneficial.

Indicators to measure help-seeking behaviour were knowing where to seek formal help, informally disclosing, seeking formal help, and receiving formal help. A total of 33.7% of AGYW and 33.1% of ABYM who had experienced sexual violence had knowledge of where to seek formal help. These numbers are slightly higher than those observed in the 2010 Violence Against Children Survey (VACS) [36]. The increase in knowledge can be attributed to stronger campaigns against sexual violence and improved response services. However, further monitoring is needed to assess how this knowledge of formal places changes over time and how these changes relate to different violence prevention efforts. 

Disclosing a case of sexual violence is an important step towards obtaining the medical, psychological and legal services needed by victims. In this study, disclosure varied by gender. AGYW reported higher rates of disclosure than ABYM (45% versus 23%). This was supported by a study in Kenya using VACS 2010 data [69]. There was an increased disclosure rate in the VACS 2019 dataset compared to the VACS 2010 dataset results for Kenya. There are several explanations for the lower rate of disclosure in ABYM compared to AGYW. The norms surrounding masculinity discourage men from disclosing sexual violence. Such norms expect men to be able to endure or cope with sexual abuse on their own [34,35]. Additionally, the stigmatization and criminalization of homosexuality in most parts of Africa [70] also limits disclosure because males fear they will be labelled as homosexual [71,72].

It is also worth noting that the disclosure rates reported in this study (45% for AGYW versus 23% for ABYM) are much lower than those observed in high-income countries [73,74,75]. For instance, in the United States, national survey data show that approximately 63% of female and 23% of male survivors disclose their experience, most often to a friend or family member [76]. Similarly, a European Union-wide survey reported that between 50% and 70% of women disclosed experiences of sexual violence, depending on the country [77]. Low disclosure rates in Kenya can be attributed to cultural/gender norms, which discourage disclosure. These norms shape attitudes towards sexual violence and result in stigma related to disclosure. Moreover, some cultural norms encourage the normalization of sexual violence, which distorts understanding and creates silence among victims [7,8,34,71]. Inadequate response and support services can also hinder reporting. In situations where adequate post-violence services are available, children and youth may be unaware of their existence, may not have the resources to access them, or worse, may not be aware that they are supposed to seek help [69]. This study also indicated that girls who received positive parenting were almost four times more likely to disclose an incident of sexual violence than girls who did not. Positive parenting is closely associated with healthy parental involvement and good communication that builds trust between parent and child and may promote disclosure [78]. In addition, positive parenting increases a child’s sense of self-efficacy so that they are better prepared to respond appropriately when exposed to sexual violence, and this includes disclosure [78].

This study has reinforced the urgency of scaling up sexual violence prevention and response programming, using the INSPIRE framework as a guiding foundation. Our findings point to critical gaps that align with INSPIRE’s core strategies, particularly the need to challenge entrenched male-favouring gender norms, promote nurturing parenting and caregiving, and invest in life skills education within schools and communities. These interventions are essential not only for prevention but also for breaking intergenerational cycles of violence. Moreover, given the low rates of formal help-seeking and disclosure observed in this study, especially among males, it is imperative to enhance awareness of, trust in, and accessibility to survivor-centred support services. Without these improvements, many survivors are likely to remain unsupported and at continued risk of further harm. However, while improving awareness and encouraging disclosure are critical, these efforts must be matched by increased investment in support infrastructure to avoid overwhelming under-resourced services. In many contexts, including some high-income settings, existing support systems already struggle to meet demand [79]. In low-resource settings, this gap may be even more pronounced. It is therefore essential to also strengthen complementary support mechanisms, including peer-led support groups [80], school-based counselling [81], referral systems within community and faith-based organizations [82,83], and legal aid initiatives [84].

A key strength of this study is that we tried as much as possible to situate the predictors before the outcomes, to attenuate temporal precedence bias. Moreover, this study utilized composite variables to measure sexual violence, positive parenting, gender inequitable attitudes, cyberbullying, and life skills training, thus yielding more robust results compared to if unidimensional variables were used. However, this study was not without limitations. First, it was not possible to assess associations between INSPIRE-aligned predictors and outcomes for males due to sample size limitations. Secondly, the cross-sectional design of VACS makes it impossible to make causal links. Third, being a household survey, our data excluded street connected and institutionalized AYP who might be at a greater risk of sexual violence than those living in households. Lastly, this study is limited by its reliance on victim-reported data, which constrains the ability to examine perpetrator-related dynamics. As such, while we identify victim-related risk and protective factors, we caution against interpreting these as implying survivor responsibility. Rather, these findings should be viewed within the ethical and analytical boundaries of a victim-centred dataset. A more comprehensive understanding of violence requires future research that incorporates perpetrator behaviour, relational dynamics, and enabling environments [85].

## 5. Conclusions

Findings from this study highlight the need to strengthen life skills training programmes in schools, as well as within communities. It would also make sense to inculcate cyberbullying in the life skills training curriculum, especially with increased use of social media by young people in recent times. More attention also needs to be directed towards parenting programmes given the role of parenting in the sexual violence cascade. In a bid to improve response and support services post-sexual violence, future research ought to look comprehensively into service-seeking to map out the kind of services available or otherwise. This study also highlights the need for sustained efforts to address gender inequitable norms. Of note, this study has demonstrated the centrality of the INSPIRE framework in sexual violence prevention programming and policy.

## Figures and Tables

**Figure 1 ijerph-22-00863-f001:**
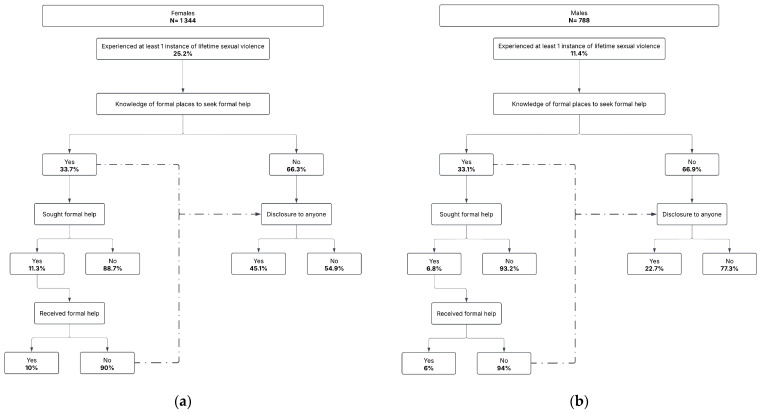
(**a**,**b**): Pathway from lifetime sexual violence experience to help-seeking among AGYW and ABYM.

**Table 1 ijerph-22-00863-t001:** Weighted descriptive statistics of sociodemographic characteristics, past-year and lifetime sexual violence forms and predictor variables for AYP.

Variables	FEMALES Weighted% (95% CI)/Mean (SD)	MALES Weighted% (95% CI)/Mean (SD)
**Sociodemographic variables**		
Age (13–24 years)	**Mean (SD)** 17.87 (3.47)	**Mean (SD)** 17.79 (3.30)
Ever married or be in a relationship	20.9 (18.3–23.7)	5.5 (4.3–7.2)
Primary education	45.3 (41.6–49.1)	41.9 (36.8–47.2)
More than Primary education	54.6 (50.8–58.4)	58.1 (52.8–63.2)
Ever tested for HIV/AIDS	35.2 (32.4–38.2)	41.9 (36.8–47.2)
Any orphanhood	23.4 (20.2–26.8)	22.0 (17.9–2.67)
Household poverty	43.1 (38.6–47.7)	36.9 (29.7–44.8)
**Predictors variables**		
Experienced cyberbullying	30.0 (26.7–33.4)	26.8 (22.1–32)
Received life skills training	20.8 (17.9–24.0)	21.1 (16.9–26.1)
Positive parental monitoring	28.1 (24.7–31.7)	16.5 (12.2–21.9)
Experienced physical violence at home	36.1 (32.4–39.9)	40.2 (34.1–46.6)
Experienced emotional violence at home	16.8 (14.4–19.5)	11.5 (8.5–15.4)
Endorsement of gender inequitable attitudes	61.8 (58.6–64.8)	68.7 (62.6–74.3)
Justification of wife beating	49.8 (45.8–53.9)	47.8 (42.2–53.4)
**Past year sexual violence forms**		
Attempted forced sex	4.0 (2.6–6.1)	2.0 (0.9–4.5)
Physically forced sex	0.8 (0.4–1.6)	0.8 (0.2–3.5)
Pressured sex	0.7 (0.4–1.0)	0.8 (0.2–3.6)
Unwanted touching	3.2 (2.2–4.7)	1.3 (0.7–2.5)
Any past year sexual violence	6.2 (4.7–8.1)	3.6 (2.2–6.0)
**Lifetime sexual violence forms**		
Attempted forced sex	15.3 (13.0–17.9)	6.8 (4.3–10.5)
Physically forced sex	5.4 (4.0–7.2)	1.8 (0.9–3.8)
Pressured sex	6.4 (5.1–8.0)	2.9 (1.5–5.4)
Unwanted touching	11.4 (8.9–14.4)	4.7 (3.0–7.1)
Any lifetime sexual violence	25.2 (22.4–28.3)	11.4 (9.1–14.1)

Note: Age is presented as a mean (SD); all other variables are reported as weighted percentages with 95% confidence.

**Table 2 ijerph-22-00863-t002:** Pathway from lifetime sexual violence experience to help-seeking for males and females.

	FEMALES	MALES	*p* Value
	Weighted% (95% CI)	Weighted% (95% CI)	
Lifetime sexual violence experience	25.2 (22.4–28.3)	11.4 (9.1–14.1)	**0.00**
Knowledge of where to seek formal help	33.7 (28.2–39.7)	33.1 (18.0–52.6)	0.85
Informal disclosure	45.1 (37.1–53.4)	22.7 (12.0–38.9)	**0.00**
Sought formal help	11.3 (7.8–16.1)	6.8 (2.3–18.5)	0.25
Received formal help	10 (6.5–15.0)	6.0 (1.8–18.5)	0.36

Note: Bolded *p* Values are statistically significant (*p* < 0.05).

**Table 3 ijerph-22-00863-t003:** Patterns of sexual violence forms (past year) among adolescent girls and young women by exposure variables.

Variables	Unwanted Touching	Attempted Forced Sex	Physically Forced Sex	Pressured Sex	Any Sexual Violence Form
	% (95% CI)	% (95% CI)	% (95% CI)	% (95% CI)	% (95% CI)
Endorsement of gender inequitable attitudes	78.4 * (64.0–88.1)	61.8 (39.7–80.0)	83.6 (51.4–96.1)	82 (46.5–96.0)	78.4 (64.0–88.1)
Justification of wife beating	48.8 *** (31.2–66.7)	70.9 ** (51.2–85.0)	33.8 (9.1–72.2)	24.8 (5.5–65.1)	48.8 (31.2–66.7)
Experienced physical violence at home	50.7 (35.0–66.3)	40.2 (22.1–61.6)	73.8 ** (35.5–93.5)	78.5 ** (39.6–95.3)	50.7 * (35.0–66.3)
Experienced emotional violence at home	36.6 *** (23.5–52.0)	53 *** (32.9–72.1)	21.4 (5.4–56.2)	21.8 (5.1–58.9)	36.6 *** (23.5–52.0)
Positive Parental monitoring	12.1 * (5.8–23.4)	14.3 (3.9–40.5)	7.3 (1.4–31.2)	_	12.1 (5.8–23.4)
Experienced cyberbullying	62.1 *** (44.6–76.9)	81.8 *** (65.0–91.6)	85.3 *** (52.4–96.8)	83.1 *** (49.6–96.1)	62.1 *** (44.6–76.9)
Life skills training	19.7 (11.1–32.7)	5.6 *** (2.0–14.5)	9.1 (1.5–40.2)	8.2 (1.0–45.2)	19.7 *** (11.1–32.7)

Confidence intervals in parentheses: *** *p* < 0.01, ** *p* < 0.05, * *p* < 0.1. The blank cells are due to less or no observations.

**Table 4 ijerph-22-00863-t004:** Logistic model predicting past year sexual violence experience among adolescent girls and young women.

Variables	Unwanted Touching	Attempted Forced Sex	Physically Forced Sex	Pressured Sex	Any Sexual Violence Form
	aOR (95% CI)	aOR (95% CI)	aOR (95% CI)	aOR (95% CI)	aOR (95% CI)
**Predictor Variables**					
Endorsement of gender inequitable attitudes	3.07 ** (1.10–8.56)	0.73 (0.22–2.40)	1.38 (0.17–10.99)	2.69 (0.23–31.38)	1.44 (0.60–3.44)
Justification of wife beating	0.52 (0.20–1.33)	2.01 (0.73–5.53)	1.65 (0.25–11.13)	0.54 (0.07–3.89)	1.24 (0.58–2.67)
Experienced physical violence at home	1.16 (0.46–2.93)	0.82 (0.36–1.90)	3.13 (0.56–17.57)	2.54 * (0.95–6.82)	1.13 (0.65–1.97)
Experienced emotional violence at home	2.36 (0.78–7.14)	2.18 (0.83–5.70)	1.34 (0.34–5.35)	3.03 (0.71–12.94)	2.11 ** (1.17–3.81)
Positive parental monitoring	0.31 ** (0.10–0.99)	0.80 (0.20–3.25)	0.20 ** (0.06–0.70)	_	0.56 (0.18–1.73)
Experienced cyberbullying	3.31 ** (1.24–8.84)	8.43 *** (2.51–28.30)	5.84 ** (1.05–32.49)	_	5.90 *** (2.83–12.29)
Life skills training	0.99 (0.45–2.16)	0.22 ** (0.07–0.73)	1.70 (0.19–14.98)	_	0.49 ** (0.26–0.92)
**Controls**					
Age	0.97 (0.84–1.1)	1.04 (0.92–1.19)	1.33 *** (1.16–1.52)	2.17 *** (1.80–2.62)	1.03 (0.93–1.14)
Ever married or be in a relationship	0.11 ** (0.02–0.69)	0.22 *** (0.08–0.66)	0.61 (0.13–2.81)	0.13 *** (0.03–0.46)	0.31 *** (0.14–0.67)
Education level (Ref: Primary)	1.05 *** (1.02–1.09)	1.04 ** (1.01–1.07)	1.04 (0.95–1.13)	0.08 *** (0.02–0.37)	1.02 (0.98–1.06)
Ever tested for HIV/AIDS	0.85 (0.27–2.61)	0.57 (0.21–1.58)	0.27 * (0.07–1.10)	_	0.89 (0.37–2.13)
Orphanhood	0.52 (0.15–1.84)	0.48 (0.17–1.34)	1.43 (0.46–4.40)	0.41 (0.07–2.25)	0.72 (0.34–1.56)
Household poverty	0.64 (0.30–1.36)	0.83 (0.34–2.05)	0.58 (0.16–2.06)	0.51 (0.18–1.44)	0.72 (0.37–1.42)

Confidence intervals in parentheses: *** *p* < 0.01, ** *p* < 0.05, * *p* < 0.1. The blank cells are due to less or no observations.

**Table 5 ijerph-22-00863-t005:** Logistic model predicting informal disclosure for lifetime sexual violence experiences among adolescent girls and young women.

Predictors	Informal Disclosure	
	aOR (95%CI)	*p*-Value
**Exposure Variables**		
Endorsement of gender inequitable attitudes	1.17 (0.57–2.42)	0.663
Justification of wife beating	0.69 (0.35–1.37)	0.284
Experienced physical violence at home	0.64 (0.35–1.19)	0.153
Experienced emotional violence at home	1.62 (0.80–3.27)	0.172
Positive parental monitoring	3.85 (1.56–9.46)	**0.004**
Experienced cyberbullying	2.13 (1.31–3.45)	**0.003**
Life skills training	0.97 (0.53–1.75)	0.911
**Control**		
Age	0.93 (0.85–1.02)	0.130

Note: Bolded *p* Values are statistically significant (*p* < 0.05).

## Data Availability

The datasets for this study are publicly available at the Together for Girls website at; https://www.togetherforgirls.org/violence-children-surveys/ (accessed on 19 November 2022).

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
