# Peer review of "Preventing Sexual Violence and Strengthening Post-Victimization Support Among Adolescents and Young People in Kenya: An INSPIRE-Aligned Analysis of the 2019 Violence Against Children Survey (VACS)"

_ijerph, 2025, doi:10.3390/ijerph22060863_

Round 1
Reviewer 1 Report
Comments and Suggestions for Authors
This study investigates two important outcomes—(1) experiences of sexual violence and (2) post-violence help-seeking behaviors—using the INSPIRE framework to identify relevant risk factors through the analysis of 2019 Kenya VACS data. Conceptually, the approach is well-grounded and highly relevant to both preventive and responsive action, particularly in terms of improving the adequacy of post-victimization services.
The practical implications of the study are also noteworthy, as they encompass both the prevention of sexual violence and the enhancement of intervention services’ accessibility and adequacy after victimization. This dual focus—on prevention and response—is consistently highlighted throughout the manuscript. In fact, the inclusion of help-seeking behaviors as one of the outcomes reinforces this dual orientation. However, it would be valuable to underline more clearly (e.g., title) that the findings also point to the need to strengthen access to post-violence response and support services, in addition to prevention.
The findings strongly support the need to address cultural and social norms that stigmatize male victims and normalize female victimization. They also highlight the role of parenting practices, both as a protective factor in preventing victimization and as a key domain in fostering disclosure and help-seeking behaviours. While the authors acknowledge the limitation of not being able to assess associations between predictors and outcomes in the male subsample due to sample size constraints, I believe it would be important to position gender more explicitly as a core analytical focus in the conceptualization and introduction of the paper. This would allow for a more in-depth gendered discussion later in the manuscript.
Relatedly, I suggest the authors more clearly justify the methodological choice of conducting separate analyses for male and female subsamples rather than controlling for gender as a covariate. A brief rationale for this choice—both theoretical and methodological—would enhance the robustness of the study design.
To improve the reader’s understanding of the sample, I recommend including sample characteristics in the abstract, particularly the sample size and age range. Similarly, Section 2.1 (Data) should descriptive information such as age range (13–24 years) and gender should be provided in this section rather than being treated as part of the results, since this is a secondary data analysis.
The manuscript refers to a “key strength” of the study as the effort to temporally situate predictors before outcomes. This is indeed a crucial issue in identifying risk factors. I suggest the authors elaborate on how this was achieved—what criteria or procedures were used to ensure predictors were temporally prior to outcomes—and whether any potential predictors were excluded on this basis.
Cyberbullying is identified as an important predictor, which is highly relevant. However, this point would benefit from deeper discussion. For instance, frameworks such as polyvictimization theory could be considered to explore the interconnections between cyberbullying and sexual violence. Are the same individuals experiencing both forms of victimization? Could the same perpetrators be involved? What patterns of risk or vulnerability can be inferred? Without careful framing, there is a risk of reinforcing victim-blaming narratives—especially if the emphasis shifts toward victim behavior patterns. Thus, this section could be strengthened by proposing hypotheses for future research that examine these intersections more critically and comprehensively.
Lastly, I suggest the discussion reflect on a broader conceptual point: the risk of overemphasizing victim-related risk factors without sufficiently considering perpetrator-related dynamics. Given that the data pertain only to victimization, it is important to acknowledge this limitation. Including such a reflection would help clarify that the intention is not to minimize offender responsibility, but rather to point out the analytical and ethical limits of a victim-only perspective.
Despite these considerations, my overall assessment of this work is highly positive. The manuscript presents valuable findings and makes a meaningful contribution to the field. Addressing the above suggestions would strengthen the study’s focus and provide a more comprehensive understanding of the risk factors for sexual violence victimization and the determinants of help-seeking, with concrete implications for both prevention and the organization of post-victimization services.
Reviewer 2 Report
Comments and Suggestions for Authors
Please find detailed comments in the attached document.

The quality of English language is mostly good, but the paper would benefit from a thorough proofread and grammatical check, as there are a few instances where it could be improved.
Reviewer 3 Report
Comments and Suggestions for Authors
This is an important topic of study.
The manuscript would be improved by adding some theoretical background related to the predictors; e.g., why would cyber-bullying be related?
For table 1 the values are not very clear - not all of the values are means under the column for means
Figure 1 does not seem necessary
There were a lot of analyses due to many criterion variables - could the authors focus on fewer, or some kind of combined variable?
Round 2
Reviewer 3 Report
Comments and Suggestions for Authors
The authors made positive changes to the manuscript suggested in the review process.